# Relationship between the honeydew of mealy bugs and the growth of *Phlebopus portentosus*

**Yi-Wei Fang**  **, Wen-Bing Wang, Ming-Xia He, Xin-Jing Xu, Feng Gao, Jing Liu, Tian-Wei Yang, Yang Cao, Tao Yang, Yun Wang\*, Chun-Xia Zhang\***

Yunnan Institute of Tropical Crops, Jinghong, Yunnan, China

\* zhangchunxia7084@163.com (CXZ); wangy10melrose@hotmail.com (YW)

## Abstract

### Background

*Phlebopus portentosus* and mealy bugs form a fungus-insect gall on the roots of host plants. The fungus and mealy bugs benefit mutually through the gall, which is the key link in the nutritional mechanism of *P. portentosus*. The cavity of the fungus-insect gall provides an ideal shelter for mealy bugs survival and reproduction, but how does *P. portentosus* benefit from this symbiotic relationship?

### Methodology and results

Anatomical examination of fungus-insect galls revealed that one or more mealy bugs of different generations were living inside the galls. The mealy bug's mouthpart could penetrate through the mycelium layer of the inside of the gall and suck plant juice from the host plant root. Mealy bugs excreted honeydew inside or outside the galls. The results of both honeydew agar medium and quartz tests showed that the honeydew can attract and promote the mycelial growth of *P. portentosus*. A test of the relationship between the honeydew and the formation of the fungus-insect gall showed that honeydew promoted gall formation.

### Conclusions

All experimental results in this study show that the honeydew secreted by mealy bugs can attract and promote the mycelial growth of *P. portentosus*, forming a fungus-insect gall, because mealy bugs' honeydew is rich in amino acids and sugars.

## 1. Introduction

*Phlebopus portentosus* (Berk. & Broome) Boedijn is a delectable wild edible fungus in the pan-tropical region of Yunnan, Panzhihua of Sichuan, southern Guangxi Prov., China. It has also been found in Thailand and Sri Lanka [1–6]. To date, *P. portentosus* is the only species in the Boletales that can produce sporocarps in culture without a host plant [2,7–8]. Yunnan Institute of Tropical Crops, Yunnan, China, has conducted research on *P. portentosus* since 2003. After

**Funding:** This study was supported by: the Key Project of Applied Basic Research of Yunnan

Province (No. 2017FA017), Youth Project of Applied Basic Research of Yunnan Province (No. 2018FD157), National Natural Science Foundation of China (No. 31560008), Funds of Sci-Tech Innovation System Construction for Tropical Crops of Yunnan Province (No. RF2019-12),and the project of Sci-Tech Talents and the Platform of Yunnan Province (No. 2019HB069).

**Competing interests:** The authors have declared that no competing interests exist.

years of research on the ecology and biotrophy of *P. portentosus*, its special biotrophical relationship with mealy bugs has been gradually revealed [4,9–10]. *P. portentosus* and mealy bugs form a fungus-insect gall on the roots of host plants. It is called a "fungus-insect gall" because the gall differs from common insect galls in that the crusty walls are made by the mycelia of *P. portentosus* rather than plant tissues [3–4,6].

The fungus (*P. portentosus*), insect (mealy bug), and plant form a unique tripartite nutritional relationship. We suspected the fungus-insect gall is the key link in this nutritional relationship, through these galls, *P. portentosus* forms symbiotic associations with mealy bugs, and the gall provides a safe and comfortable living environment for the mealy bugs. In turn, a large amount of honeydew secreted by the mealy bugs provides the necessary nutrients for the growth of *P. portentous* [2–4]. Preliminary chemical analysis of 100 g of mixed honeydew produced by *Dysmicoccus neobrevipes* and *Crisiococcus matsumotoi* (Shiraiwa) contained 6.1 g of hydrolyzed amino acids, 3.9 g of free amino acids with 17 kinds of amino acids, [4]. Currently, the fungus-insect gall has been found on the roots of 21 plant species, and 13 mealy bug species are associated with the fungus. Among them, 11 species belong to the family Pseudococcidae, while the other two belong to Monophlebidae and Eriococcidae [3–4].

As early as the 1940s, Gonçalves et al. (1941) reported that *Phlebopus tropicus* formed similar galls (known as "crypta") with *Pseudococcus comstocki* on the roots of citrus plants in citrus orchards in Brazil [11]. The honeydew excreted by the *P. comstocki* attracted ants, which in turn helped the insects to grow better, finally causing extinction of the citrus cultivation in Brazil due to *P. comstocki* mass reproduction. Brundrett and Kendrick [12] found that another member of Boletinellaceae, *Boletinellus merulioides* (Schweinitz) Murrill, also formed similar galls (sclerotia) on the roots of *Fraxinus* trees [12]. Recently, Lumyong et al. [5] and Kumla et al. [8] reported that *P. portentosus* formed galls with *Paraputo banzigeri* Williams on the roots of *Dimocarpus longan* Lour [5, 8]. Singer [13] and Watling [14] assumed that many members of *Phlebopus* might form some kinds of symbiotic associations with root aphids [13–14]. It has been reported that mealy bugs' honeydew is rich in sugars, amino acids, niacinamide, protein, mineral elements and vitamin B [15–18], which may support fungal growth. However, the nutritional relationship between the fungus and mealy bugs, nor its mechanism, has been specified. This article is a preliminary summary of years of research on the relationship between honeydew secreted by mealy bugs and the growth of *P. portentosus*.

## 2. Materials and methods

### 2.1. Anatomical examination of the fungus-insect gall

Samples of fungus-insect galls were collected from the roots of *Eriobotrya japonica (Thunb.) Lindl.* in the Mount happy Loquat Garden (24°06′N, 99°54′E), Lincang County, Yunnan, China, on 5 November 2015, and the roots of *Wedelia chinensis* (Osbeck.) Merr. in Mount Xinghuoshan (22°07′N, 100°11′E), Jinghong, Yunnan, China, on 27 November 2015, separately. Mount Xinghuoshan is not national park or other protected area of land, where permission was not required for taking samples. These field studies did not involve endangered or protected species. These samples were wrapped with moistened tissue paper and brought into the laboratory. The fungus-insect galls were cut longitudinally into small sections of 3–4 cm in length, then kept individually in petri dishes with moistened tissue paper underneath. These sections were examined under a stereomicroscope (LEICA M125).

### 2.2. Raising honeydew

Mealy bugs (*Dysmicoccus neobrevipes* (Beardsley)) were collected from fungus-insect galls of the roots of *Delonix regia* at the experimental base of the Yunnan Institute of Tropical Crops

on 28 October 2015. The mealy bugs were raised on a pumpkin (*Cucurbita moschata*) surface at a room temperature of 26 ± 2˚C in a laboratory. A large amount of honeydew accumulated as the mealy bug numbers increased. The honeydew was transferred into centrifuge tubes with a micro-syringe and stored at -20˚C.

## 2.3. Effects of honeydew on the growth of *P. portentosus*

**2.3.1. Strain and culture medium.** Strain 17016 was isolated from the tissue of a fruiting body of *P. portentosus* collected under a tree of *D. regia* in Gadong Town, Jinghong, Yunnan, China, and was used for inoculation. The culture method of inoculation was performed according to previously described methods [3].

**2.3.2. Experiment 1 on agar medium.** Strain 17016 was cultured in petri dishes with potato dextrose agar (PDA) and incubated at 28˚C for 20 days. Three agar medium were used to test the effects of honeydew on mycelial growth. One was a 1.0% honeydew agar medium, which was made by adding 1.0% honeydew to a 2.0% agar medium. The second was PDA medium as the positive control. The third was made from 2.0% pure agar. All media were autoclaved at 120˚C for 20 mins, and then 15.0 ml of each was poured into separate petri dishes (BS-90-D, Japan). Each was inoculated with a 5 mm radius disc of 17016 PDA mycelial plugs. All inoculated mycelial plugs were incubated in a dark room with 28˚C ± 2˚C and a relative humidity of 60% ± 5%.

The colony diameters were measured with a venire caliper (by the cross method) on the 14th day after inoculation. The average growth rate were calculated and expressed with standard errors. The experimental data were analyzed by SPSS 23 for the least significance differential (LSD) test. At the same time, colony growth features, such as the mycelium density, colony color, and colony edge characteristics, were observed and recorded.

**2.3.3. Experiment 2 on quartz pebbles.** Preparation of the fungal solid substrate of strain 17016 followed the method of Ji et al. [2]. 120 glass jars (5 cm×5 cm×9 cm) were filled with 100 g of solid substrate, sealed with sealing films and sterilized at 121˚C for 90 mins. All the jars were inoculated with the solid substrate of strain 17016 in a dark room at 28˚C ± 2˚C and a relative humidity of 60% ± 5% for 18 days.

Quartz pebbles (0.2–0.3 mm) were autoclaved at 121˚C for 20 mins. The honeydew was autoclaved at 110˚C for 5 mins. 600 pebbles were smeared with the honeydew under aseptic conditions. Ten honeydew pebbles per jar were placed on the solid substrate surface in 60 jars. Ten normal quartz pebbles with sterilized water per jar were placed in the other 60 jars as a control. All jars were sealed with a sealing film and incubated in a dark incubator at 28˚C ± 2˚C and a relative humidity of 60% for 14 days.

## 2.4. Relationship between honeydew and fungus-insect gall formation

Preparation of mealy bug adults of *D. neobrevipes* was described in section 2.2. Preparation of the fungal solid substrate of strain 17016 followed the method of Ji et al. [2]. Preparation of the fungal solid substrate jars was described in section 2.3.3. 120 jars were prepared. Ten heads of mealy bug adults were picked up and placed on the surface of the solid substrate in each jar. 60 jars received mealy bug adults as the test group. The other 60 jars received pebbles (0.2–0.3 mm) on the surface of the solid substrate as a control. The pebbles were autoclaved at 121˚C for 20 mins. Ten pebbles were used for each jar. All jars were inoculated in a dark room at 28˚C ± 2˚C and a relative humidity of 60% ± 5%.

The formation of the fungus-insect gall in three jars was examined and recorded every three days. The numbers and sizes of the galls formed were recorded. The inside of each gall was examined under a stereomicroscope (LEICA M125) to determine its survival.

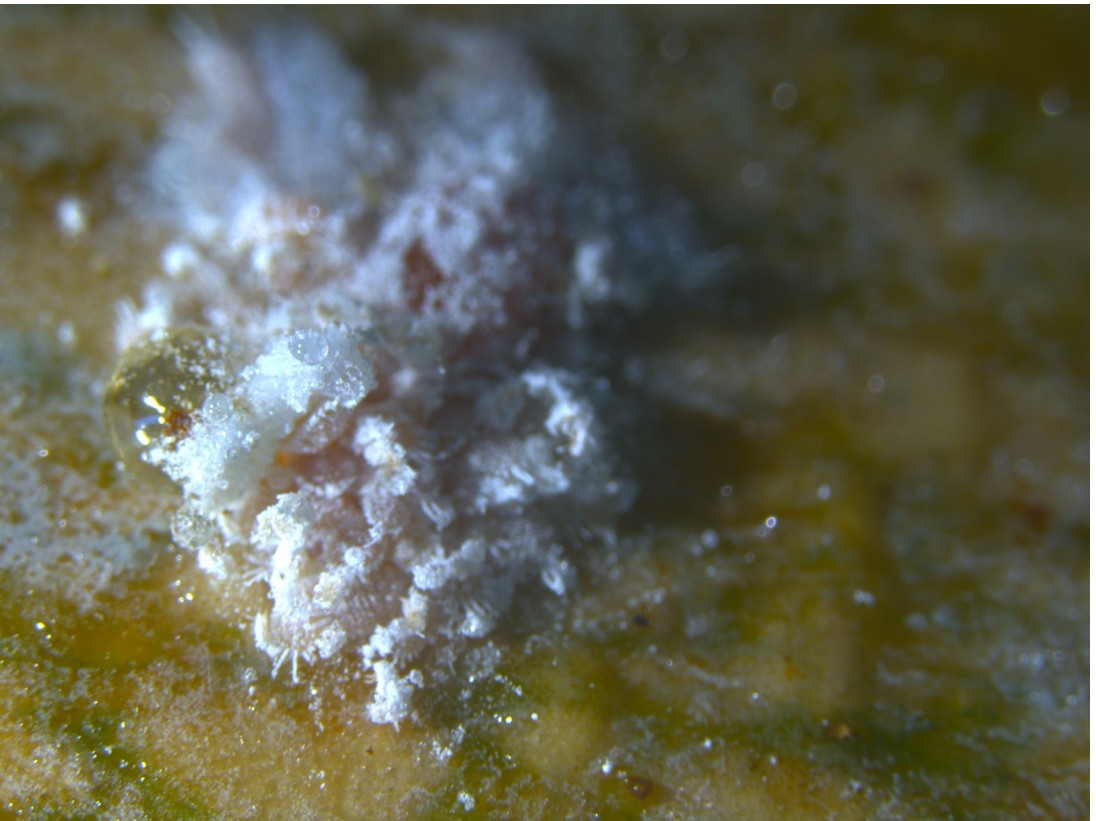

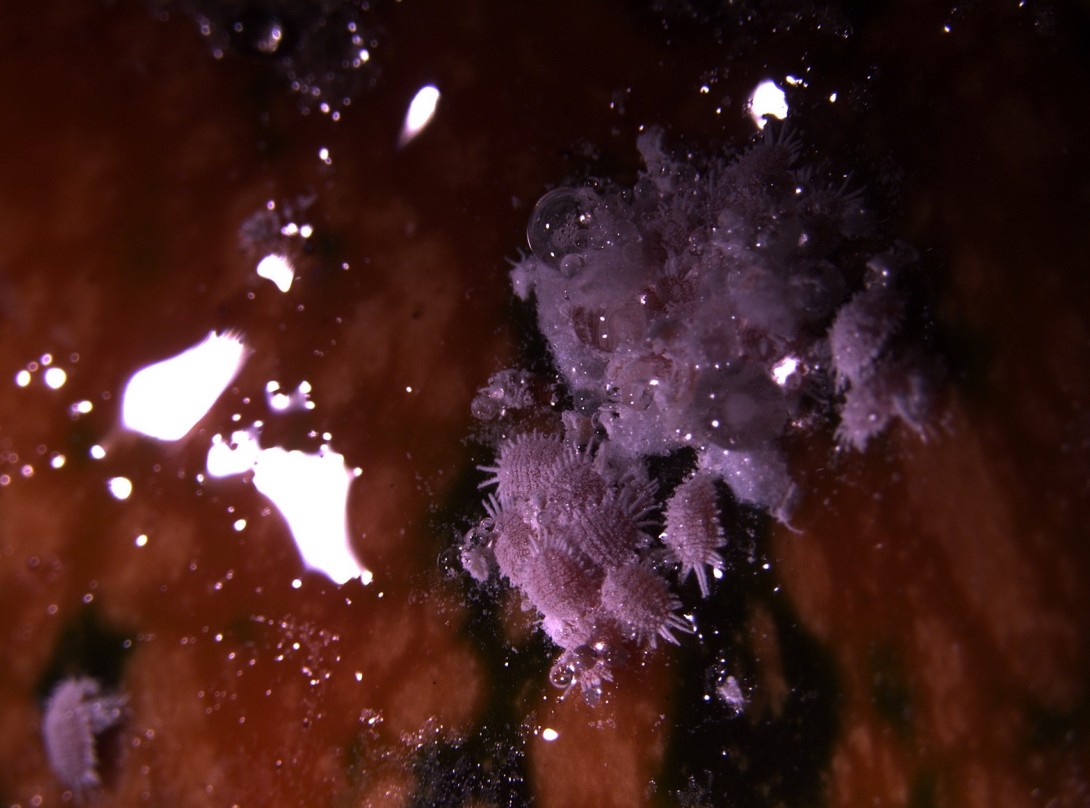

**Fig 1. Mass reproduction of *D. neobrevipes* and accumulation of a large amount of honeydew.**

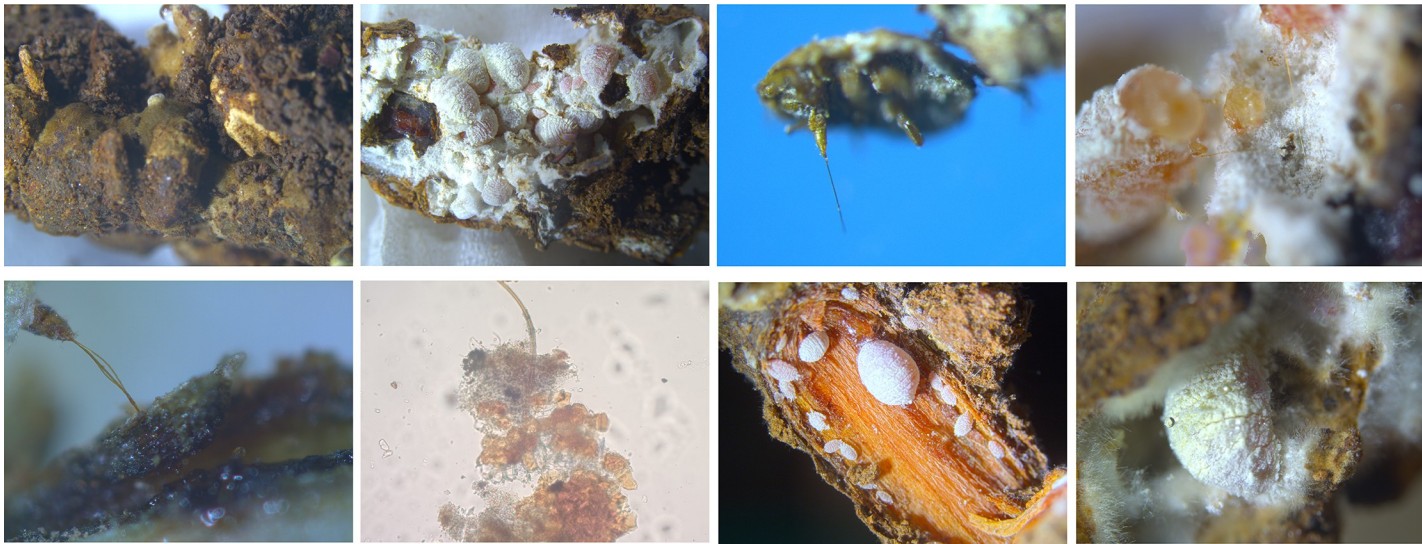

**Fig 2. Anatomical examination of the fungus-insect gall.**

## 3. Results

### 3.1. Raising honeydew

About 20 days, a large amount of honeydew had accumulated on the pumpkin surface as a by-product of mealy bug (*D. neobrevipes*) mass reproduction (Fig 1A and 1B).

### 3.2. Anatomical examination of the fungus-insect gall

One or more mealy bugs of different generations were living inside the galls (Fig 2A and 2B). The mealy bug's thin, long, flexible mouthpart (Fig 2C) penetrated through the mycelium layer inside the gall and extracted plant juice from the host plant root (Fig 2D and 2E). When the fungus-insect gall was destroyed, the mealy bugs moved on the cortex of roots to absorb nutrients (Fig 2G and 2F). Mealy bugs can excrete honeydew inside or outside the galls (Fig 2H).

### 3.3. Effects of honeydew on the growth of *P. portentosus*

**Experiment 1 on agar medium.** Significant differences in colony growth between the medium with 1% honeydew and the pure agar medium were observed (Table 1). On the 1% honeydew medium, a normal colony developed with dense, thick and sturdy mycelium within 14 days (Fig 3A). The colony growth rate was 3.87 ± 0.08mm/day. However, on the agar

**Table 1. Mycelial growth on three media.**

| Medium | Fungal mycelia | Sclerotium | Color | Growth rate Mean ± SE (mm/d) |
|---|---|---|---|---|
| Pure agar | Very thin and sparse | No | Yellow brown | 2.52 ± 0.34 b |
| Honeydew | Dense and thick | No | Yellow brown | 3.87 ± 0.08 a |
| PDA | Very dense and thick | Developed | Dark brown | 4.28 ± 0.16 a |

[a] Results are means ± SE of five replicates. Data with different letters within the same column indicates a significant difference at $P < 0.05$ according to the LSD multiple comparison test.

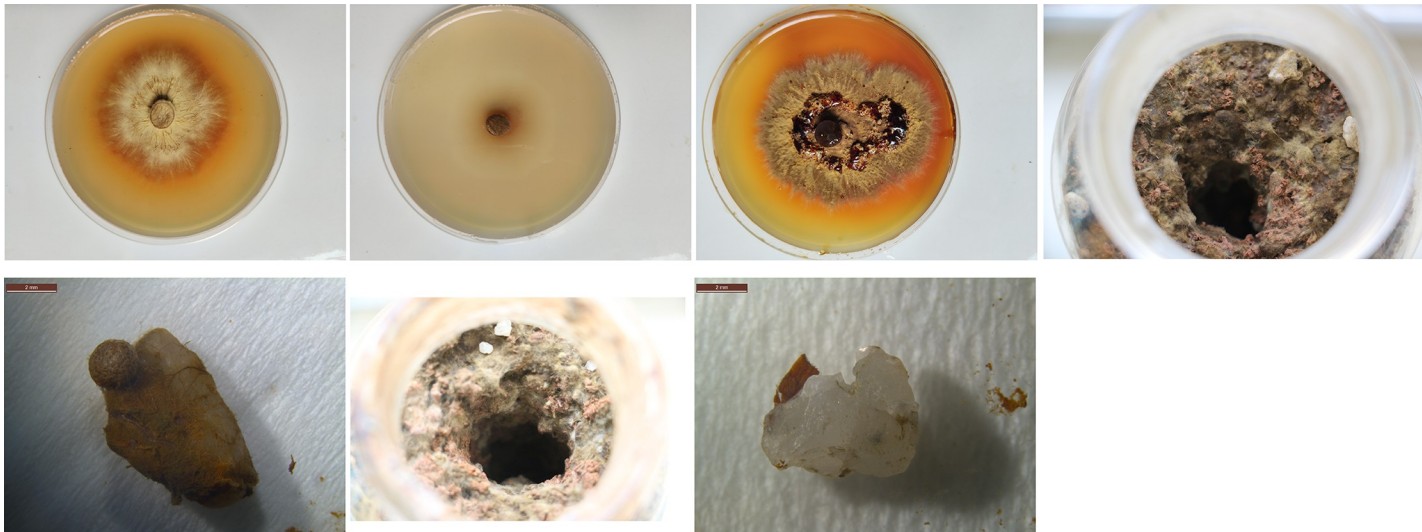

**Fig 3. Effects of honeydew on the growth of *P. portentosus*.**

medium, almost no new mycelium developed (Fig 3B). Regarding the growth rate, the fungal mycelium growth on the honeydew medium was not significantly different from that on the PDA (4.28 ± 0.16 mm/day) (Table 2). However, the mycelia on the PDA were denser and thicker than those on the 1% honeydew medium (Fig 3C). The experimental results showed that the honeydew could provide nutrients, including amino acids and sugars, for normal *P. portentosus* mycelium growth.

**Experiment 2 on quartz pebbles.** All quartz pebbles with honeydew of mealy bugs were colonized by the fungus. Some of them were even completely wrapped by the mycelium (Fig 3D). Sclerotia sometimes formed on their surfaces (Fig 3E). The pebbles in the control group

**Table 2. The formation of fungus-insect galls.**

| Days after the insect was placed on the solid substrate | Survival of mealy bugs (%) | Size of the galls (mm) | | TThickness of the galls |
|---|---|---|---|---|
| | | Length | Width | |
| 3 | 100% ± 0% | | | |
| 6 | 100% ± 0% | 4.55 ± 0.16 a | 2.98 ± 0.62 b | 0.18 ± 0.042 b |
| 9 | 100% ± 0% | 4.90 ± 0.62 a | 3.00 ± 0.34 b | 0.32 ± 0.039 a |
| 12 | 100% ± 0% | 5.89 ± 0.70 a | 4.08 ± 0.29 a | 0.35 ± 0.074 a |
| 15 | 86.67% ± 8.82% | 4.99 ± 0.62 a | 3.58 ± 0.23 ab | 0.29 ± 0.044 ab |
| 18 | 83.33% ± 6.67% | 5.22 ± 0.50 a | 3.58 ± 0.16 ab | 0.26 ± 0.029 ab |
| 21 | 73.33% ± 3.33% | 5.30 ± 0.32 a | 3.93 ± 0.48 ab | 0.33 ± 0.026 a |
| 24 | 23.33% ± 14.50% | 4.93 ± 0.24 a | 3.08 ± 0.23 ab | 0.35 ± 0.062 a |
| 27 | 10.00% ± 5.77% | 5.59 ± 0.25 a | 3.80 ± 0.36 ab | 0.29 ± 0.054 ab |
| 30 | 6.67% ± 3.33% | 4.91 ± 0.65 a | 3.87 ± 0.30 ab | 0.34 ± 0.041 a |
| 33 | 3.33% ± 3.33% | 4.62 ± 0.46 a | 3.70 ± 0.37 ab | 0.35 ± 0.034 a |
| 36 | 3.33% ± 3.33% | 5.46 ± 0.30 a | 3.88 ± 0.41 ab | 0.37 ± 0.021 a |
| 39 | 0% ± 0% | 4.97 ± 0.33 a | 3.35 ± 0.06 ab | 0.33 ± 0.074 a |
| 42 | 0% ± 0% | 5.22 ± 0.48 a | 3.62 ± 0.49 ab | 0.30 ± 0.011 ab |

Data with different letters within the same column indicates a significant difference at $P < 0.05$ according to the LSD multiple comparison test.

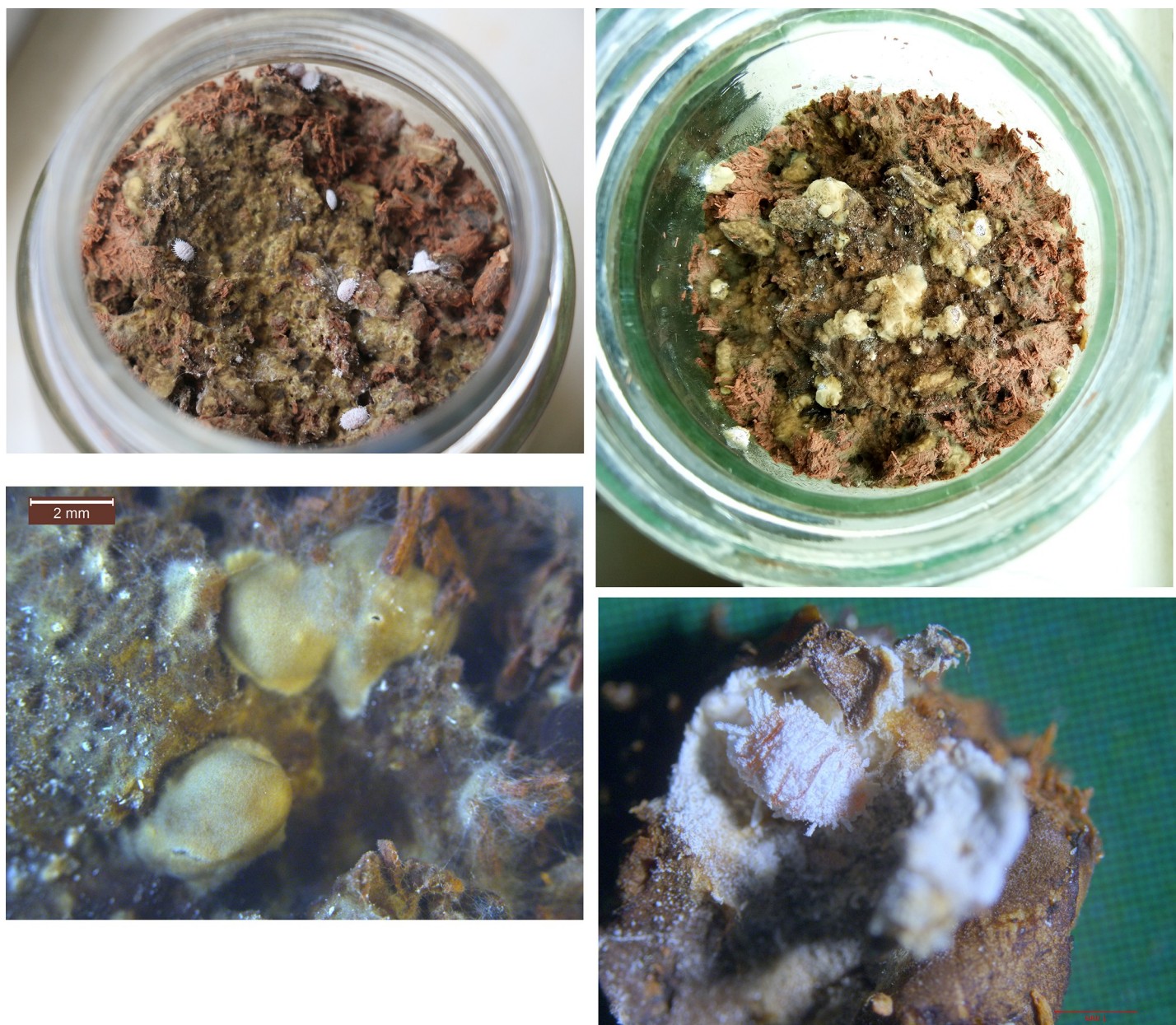

**Fig 4. Relationship between honeydew and gall formation.**

showed adherence only by very thin mycelium on the side, which was attached to the surface of the solid substrate (Fig 3F and 3G). The results indicated that the honeydew of mealy bugs attracted and promoted fungal mycelium growth.

### 3.4. Relationship between honeydew and fungus-insect gall formation

When the mealy bugs (*D. neobrevipes*) were placed on the surfaces of the solid substrate, the fungal mycelium started to produce galls covering the mealy bugs (Fig 4A). On the 6th day, the mealy bug bodies were almost completely enwrapped by the fungal mycelium (Fig 4B). On the 9th day, the galls had completely formed (Fig 4C). The sizes of the galls were quite even,

namely, 4–5 × 3–4 mm, with a wall that was 0.1 to 0.3 mm thick (Table 2). The control had no galls present.

The adult mealy bugs inside the galls gradually lost weight with a gradually reducing survival rate until the 39th day after were introduced (Fig 4D). The nymphs produced by the female mealy bugs inside the galls could survive for only 2–3 days.

## 4. Conclusions and discussion

All the results of these experiments in this study show that the honeydew secreted by mealy bugs can attract and promote the mycelium growth of *P. portentosus* and form a fungus-insect gall because the honeydew is rich in amino acids and sugars [4]. The honeydew can provide necessary nutrients for normal fungal growth, and it plays a key role in the symbiotic association between *P. portentosus* and mealy bugs.

In the agar medium test on 1 liter of 1% honeydew agar medium, the mycelium growth was not as good as that on the PDA medium, as 1 liter of 1% honeydew with 10g of honeydew only containing little sugars and free amino acids, compared with 1 liter of PDA, containing 20 g of glucose and extraction from 200 g of potato containing high proteins, fibers, potassium, vitamins [19]. A higher honeydew agar medium concentration, such as 3% or higher, could produce better colonies than those presented here, such as those seen on the PDA. Unfortunately, we could not obtain enough honeydew to test higher honeydew contents. Higher honeydew contents than those presented in this study are currently being tested in our research program.

The mealy bugs inside the galls that formed on the solid substrate eventually died due to starvation, as they could not obtain enough nutrients from the solid substrate. This study indicates that mealy bugs can be completely protected by the fungus through the gall but cannot obtain any nutrients from it. In the field, fungi forms galls on the roots of plants to shelter insects, of which the mouthparts can penetrate cortex the extract juice from the roots. In return, the mealy bugs produce abundant honeydew to nourish the fungus.

## Supporting information

**S1 Appendix. Raw data of Table 1: Mycelial growth on three media (https://doi.org/10.1371/journal.pone.0233710).**
(XLS)

**S2 Appendix. Raw data of Table 2: The formation of fungus-insect galls (https://doi.org/10.1371/journal.pone.0233710).**
(XLS)

## Author Contributions

**Conceptualization:** Yi-Wei Fang, Wen-Bing Wang, Feng Gao, Jing Liu, Tian-Wei Yang, Yang Cao, Yun Wang, Chun-Xia Zhang.

**Data curation:** Yi-Wei Fang, Chun-Xia Zhang.

**Formal analysis:** Yi-Wei Fang, Feng Gao, Tian-Wei Yang, Yun Wang, Chun-Xia Zhang.

**Funding acquisition:** Yi-Wei Fang, Ming-Xia He, Feng Gao, Yun Wang, Chun-Xia Zhang.

**Investigation:** Yi-Wei Fang, Wen-Bing Wang, Ming-Xia He, Xin-Jing Xu, Feng Gao, Jing Liu, Tian-Wei Yang, Yun Wang, Chun-Xia Zhang.

**Methodology:** Yi-Wei Fang, Ming-Xia He, Xin-Jing Xu, Feng Gao, Jing Liu, Tian-Wei Yang, Yang Cao, Tao Yang, Yun Wang.

**Project administration:** Yi-Wei Fang, Yun Wang, Chun-Xia Zhang.

**Resources:** Yi-Wei Fang, Wen-Bing Wang, Ming-Xia He, Xin-Jing Xu, Jing Liu, Tian-Wei Yang, Yang Cao, Tao Yang, Chun-Xia Zhang.

**Software:** Yi-Wei Fang, Yang Cao, Chun-Xia Zhang.

**Supervision:** Yi-Wei Fang, Wen-Bing Wang, Tao Yang, Chun-Xia Zhang.

**Validation:** Yi-Wei Fang, Xin-Jing Xu.

**Visualization:** Yi-Wei Fang.

**Writing – original draft:** Yi-Wei Fang, Yun Wang, Chun-Xia Zhang.

**Writing – review & editing:** Yi-Wei Fang, Yun Wang, Chun-Xia Zhang.

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
