## [Decision Letter · Decision Letter 0]

6 Mar 2020

PONE-D-19-30898

Relationships between the honeydew of mealy bugs and the growth of *Phlebopus portentosus*

PLOS ONE

Dear Mrs Fang,

Thank you for submitting your manuscript to PLOS ONE. After careful consideration, we feel that it has merit but does not fully meet PLOS ONE’s publication criteria as it currently stands. Therefore, we invite you to submit a revised version of the manuscript that addresses the points raised during the review process.

We would appreciate receiving your revised manuscript by Apr 20 2020 11:59PM. To enhance the reproducibility of your results, we recommend that if applicable you deposit your laboratory protocols in protocols.io, where a protocol can be assigned its own identifier (DOI) such that it can be cited independently in the future. For instructions see: http://journals.plos.org/plosone/s/submission-guidelines#loc-laboratory-protocols

We look forward to receiving your revised manuscript.

Kind regards,

Yulin Gao

Academic Editor

PLOS ONE

Review Comments to the Author

Reviewer #1: Comments to the Author

The manuscript (PONE-D-19-30898) entitled “Relationships between the honeydew of mealy bugs and the growth of Phlebopus portentosus” has been processed. The results suggest that honeydew secreted by mealy bugs can attract and improve the mycelial growth of P. portentosus and form a fungal-insect gall. It will be useful for understand how P. portentosus benefit from this symbiotic relationship. The detailed comments and suggestions are as follows.

1. In the whole passage, “fungus-insect” should be changed to “fungal-insect”.

2. Line 30, “agar medium” but not “agar media”.

3. Line 36, 56 and 199, “sugar” should be changed to “sugars”.

4. Line 40, “fungi” should be changed to “fungus”.

5. Line 72-73, the singular and plural should be consistent in this sentence.

6. Line 95, “was” should be deleted.

7. Line 102, “by” should be deleted.

8. Line 214, “a fungi” rather than “a fungus”.

9. Whole numbers less than ‘10’ should be written as words rather than numbers, such as line 147-148. In return, when numbers more than ‘10’ should be written as numbers rather than word, such as line 140.

10. The language should be further improved.

Reviewer #2: Your manuscript submission is interesting and contains information that our readers would find useful.  Unfortunately, the quality of the language is not up to the standards of the journal. 

I suggest you may wish to consider having your paper professionally edited for English language by a native English speaker and/or a professional language editing service here before resubmitting this manuscript.

and also. all the Figures should improved and make it more clear..

6. PLOS authors have the option to publish the peer review history of their article (what does this mean?). If published, this will include your full peer review and any attached files.

Reviewer #1: No

Journal Requirements:

"This study was supported by the Key Project of Applied Basic Research of Yunnan Province (No. 2017FA017), Youth Project of Applied Basic Research of Yunnan Province (No. 2018FD157), National Natural Science Foundation of China (No. 31560008), Funds of Sci-Tech Innovation System Construction for Tropical Crops of Yunnan Province (No. RF2019-12), the project of Sci-Tech Talents and the Platform of Yunnan Province (No. 2019HB069). This study was also supported by the New Zealand Institute for Plant & Food Research Ltd."

"Include this sentence at the end of your statement: The funders had no role in study design, data collection and analysis, decision to publish, or preparation of the manuscript."

"This study was supported by the Key Project of Applied Basic Research of Yunnan Province (No. 2017FA017), Youth Project of Applied Basic Research of Yunnan Province (No. 2018FD157), National Natural Science Foundation of China (No. 31560008), Funds of Sci-Tech Innovation System Construction for Tropical Crops of Yunnan Province (No. RF2019-12), the project of Sci-Tech Talents and the Platform of Yunnan Province (No. 2019HB069). This study was also supported by the New Zealand Institute for Plant & Food Research Ltd."

We note that you received funding from a commercial source: Plant & Food Research Ltd.

---

## [Author Response · Author response to Decision Letter 0]

18 Apr 2020

Response to the reviewer

Dear editor:

Thank you very much for giving us an opportunity to revise our manuscript. We appreciate the editor and reviewers very much for their constructive comments and suggestions on our manuscript (PONE-D-19-30898) entitled Relationships between the honeydew of mealy bugs and the growth of Phlebopus portentosus. 

Those comments are very helpful for revising and improving our paper, as well as the important guiding significance to other research. We have studied the comments carefully and made corrections which we hope meet with approval. The main corrections are in the manuscript and the responds to the reviewers’ comments are as follows:

Reviewer #1:

1. In the whole passage, “fungus-insect” should be changed to “fungal-insect”.

Answer: The word of “fungus-insect” comes from reference of “Zhang CX, He MX, Cao Y, Liu J, Gao F, Wang WB, et al. Fungus-insect gall of Phlebopus portentosus. Mycol. 2015; 107(1):12-20. PMID:25344264.”. In order to be consistent with our previous research, so we suggested remain “fungus-insect” instead of “fungal-insect.

2. Line 30, “agar medium” but not “agar media”.

Answer: Modified throughout the text according to the comment (Line 29).

3. Line 36, 56 and 199, “sugar” should be changed to “sugars”.

Answer: Modified throughout the text according to the comment (Line 35,58,75,198). 

4. Line 40, “fungi” should be changed to “fungus”.

Answer: Modified throughout the text according to the comment (Line 38). 

5. Line 72-73, the singular and plural should be consistent in this sentence.

Answer: Modified throughout the text according to the comment, (Line 74).

6. Line 95, “was” should be deleted.

Answer: Modified throughout the text according to the comment, “was” has been deleted. (Line 101).

7. Line 102, “by” should be deleted.

Answer: Modified throughout the text according to the comment, “by” has been deleted. (Line 108).

8. Line 214, “a fungi” rather than “a fungus”.

Answer: Modified throughout the text according to the comment (Line 213).

9. Whole numbers less than ‘10’ should be written as words rather than numbers, such as line 147-148. In return, when numbers more than ‘10’ should be written as numbers rather than word, such as line 142. 

Answer: Modified throughout the text according to the comment, whole numbers less than ‘10’ should be written as words rather than numbers,3 has been changed “three”, (Line 148-149), when numbers more than ‘10’ should be written as numbers rather than word, “one hundred twenty” changed 120 (line 125).

10. The language should be further improved.

Answer: I've asked an expert in New Zealand to edit it for me and sent to AJE Customer Service (language polishing company)to edit to make it as understandable as possible.

11. In the introduction of this article, I added “To date, P. portentosus is the only species in the Boletales that can produce sporocarps in culture without a host plant [2,7-8]”, And adjust part word order in order to better introduction our study.

Replies to Reviewer 2

Specific Comments

Comment 1: Your manuscript submission is interesting and contains information that our readers would find useful. Unfortunately, the quality of the language is not up to the standards of the journal. 

I suggest you may wish to consider having your paper professionally edited for English language by a native English speaker and/or a professional language editing service here before resubmitting this manuscript.

and also. all the Figures should improved and make it more clear.

Answer: I've asked an expert in New Zealand to edit it for me and sent to AJE Customer Service (language polishing company)to edit to make it as understandable as possible. All the Figures improved through Preflight Analysis and Conversion Engine (PACE) digital diagnostic tool to make it more clear. 

Response to the Journal Requirements

1) It appears that your ORCiD iD has not been validated in your Editorial Manager account and we are unable to proceed until that step is complete. 

Answer: I have Registered ORCiD iD.

2-3) I have removed any funding-related text from the manuscript. 

4) Please amend your list of authors on the manuscript to ensure that each author is linked to an affiliation.

Answer: I have amend list of authors on the manuscript to ensure that each author is linked to an affiliation.

5) To comply with PLOS ONE submissions requirements for field studies, please provide the following information in the Methods section of the manuscript and in the “Ethics Statement” field of the submission form (via “Edit Submission”).

Answer: Samples of fungus-insect galls were collected from the roots of Eriobotrya japonica (Thunb.) Lindl. in the Mount happy Loquat Garden (24°06′N, 99°54′E), Lincang County, Yunnan, China, and the roots of Wedelia chinensis (Osbeck.) Merr. in Mount Xinghuoshan (22°07′N, 100°11′E), Jinghong, Yunnan, China. Mountain happy Loquat Garden as experimental Location of Yunnan Institute of Tropical Crops. Mount Xinghuoshan is not national park or other protected area of land. These field studies did not involve endangered or protected species. Mount Xinghuoshan is public land, which permission to take samples from was not required.

6)I have removed the file "ZVZG33FW-sub-editing-summary.pdf".

7) This study was supported by: the Key Project of Applied Basic Research of Yunnan Province (No. 2017FA017), Youth Project of Applied Basic Research of Yunnan Province (No. 2018FD157), National Natural Science Foundation of China (No. 31560008), Funds of Sci-Tech Innovation System Construction for Tropical Crops of Yunnan Province (No. RF2019-12),and the project of Sci-Tech Talents and the Platform of Yunnan Province (No. 2019HB069).

I have deleted the New Zealand Institute for Plant & Food Research Ltd.

I confirmed that the above statements are both true and correct for our work.

8)I add “Mount Xinghuoshan is not national park or other protected area of land where permission was not required for taking samples. These field studies did not involve endangered or protected species” to the Methods section of manuscript.

---

## [Decision Letter · Decision Letter 1]

12 May 2020

Relationships between the honeydew of mealy bugs and the growth of *Phlebopus portentosus*

PONE-D-19-30898R1

Dear Dr. Fang,

We are pleased to inform you that your manuscript has been judged scientifically suitable for publication and will be formally accepted for publication once it complies with all outstanding technical requirements.

With kind regards,

Yulin Gao

Academic Editor

PLOS ONE

---

## [Editor Report · Acceptance letter]

28 May 2020

PONE-D-19-30898R1 

Relationship between the honeydew of mealy bugs and the growth of Phlebopus portentosus 

Dear Dr. Fang:

I am pleased to inform you that your manuscript has been deemed suitable for publication in PLOS ONE. Congratulations! Your manuscript is now with our production department. 

With kind regards,

on behalf of

Dr. Yulin Gao 

Academic Editor

PLOS ONE